# Local Integration as a Durable Solution? Negotiating Socioeconomic Spaces between Refugees and Host Communities in Rural Northern Uganda

Sarah Khasalamwa-Mwandha 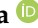

Ruralis-Institute for Rural and Regional Research, University Centre, 7491 Trondheim, Norway; khasalamwa@gmail.com or skm@ruralis.no

**Abstract:** With a growing number of displaced people, there is a need for robust approaches to coping with displacement. Uganda has a progressive refugee policy that promotes freedom of movement and the socioeconomic rights of the refugees. Specifically, refugees are often allocated land to settle and cultivate rural settlements, and the integrated social service provision facilitates interaction with host communities. However, there remain challenges in creating sustainable livelihoods for refugees in rural settlements. There exist significant tensions over shared resources such as land, water, woodlots, and grazing areas. Based on a survey of 416 households and key informant interviews with South Sudanese refugees in selected settlements in the Adjumani district, the paper highlights refugees' access to social and economic spaces as critical pathways to sustainable livelihoods and integration. Uganda's progressive policy expands the opportunity space; however, refugees still encounter significant barriers in accessing the socioeconomic spaces.

**Keywords:** refugees; sustainable livelihoods; local integration; rural transformation; protracted displacement durable solutions; northern Uganda

## 1. Introduction

Displacement and migration are significant drivers of processes of rural restructuring and transformation. Specifically, rural transformation entails the reorganisation of society in each space through economic diversification, urbanisation, social structure, and demographic growth [1] Rural transformation is often characterised by profound economic, demographic, cultural, and environmental changes [2]. In Sub-Saharan Africa, the rural spaces are changing at an unprecedented pace; however, the effects of these changes on the lives and livelihoods of rural people are under-researched, especially the changes at the micro-level of communities or households [3,4]. The changing demographic structure and livelihoods stemming from displacement and migration of refugees into rural communities is a central facet of rural transformation investigated in this paper. Rural communities in northern Uganda are not alien to the impacts of displacement and the hosting of refugees and internally displaced persons (IDPs) [5]. The refugees and host communities in northern Uganda have suffered multiple displacements spanning several decades. Refugee settlements are a prominent feature of the rural areas in northern Uganda, hosting over a million refugees. The districts of Yumbe, Adjumani, Obongi, Lamwo, and Koboko are home to over 500,000 refugees from South Sudan. In Adjumani district, where this study was conducted, there are almost as many refugees (230,101) as the host population (237,400) (UNHCR 2021 Country—Uganda (unhcr.org)). This influx of refugees is creating new communities and new ways of living in the rural areas of northern Uganda. This demographic shift has significant implications for access to resources for both the refugees and host communities.

Many refugee hosting countries face the dilemma of how and where to resettle refugees and migrants. In the global north, resettlement programs have focused on placing refugees in major urban cities. However, there are new trends and initiatives to attract refugees to

settle in the rural areas as a strategy to harness opportunities in the agricultural and manufacturing sectors and address stagnant population growth and labour shortages [6–11]. Refugee dispersal policies have emphasised resettlement in peripheral and rural areas due to the growing challenge of depopulation and labour shortages as young people migrate to urban areas [12,13]. Therefore, the resettlement of refugees in rural areas is an attempt to support rural community revival in the global north [14,15]. In Africa, resettlement of refugees in rural areas has been a default trend for many countries, including Uganda [16,17]. The significant factors determining resettlement include local absorption capacity regarding jobs, water, food, social services, ethnic and cultural compatibility, and security threats associated with refugee presence [18,19].

In Sub-Saharan Africa, there is a high dependence on humanitarian assistance due to limited economic opportunities for refugees [20–22]. Often, refugees are required to manoeuvre their survival strategies as a pathway to self-reliance [23–26]. As a result, most refugees feel obliged to stay in settlements to access humanitarian assistance. However, the enduring ruptures in their lives and livelihoods are inadequately assessed and understood [26]. Broadly, the paper investigates the diverse meanings and experiences of displacement for refugees in rural settlements in Adjumani district in northern Uganda. Specifically, the interactions between refugees and their host communities and how these impact access to resources and the socioeconomic spaces are analysed. Additionally, the paper investigates how the demographic shift caused by the influx of refugees shapes the socioeconomic interactions with the host communities.

In defining rural spaces, the significant aspects considered include the size of the population, remoteness, the nature of the economic activity, and the level of infrastructure development. The context in Adjumani deviates from the classic description of the rural [1,27], in that the refugee settlements have high population densities, but still agriculture is a significant economic activity, and a large percentage of the population is dependent on agriculture for their livelihoods. Some of the settlements have developed into informal cities. The demographic change caused by the influx of refugees in northern Uganda has triggered processes of rural transformation and restructuring manifested through the restructuring of the socio-economic morphology; territorial spatial patterns, including changes in the demographic structures, employment opportunities, community organisation, lifestyles and standards of living; accessibility; rural culture; rural industries and rural production; and living and ecological spaces [27–29]. The data on which the analysis is based were collected in two phases, July 2017 and September 2020. The data were comprised of a household survey, key informant interviews, and focus group discussions. The paper is set out as follows. First, the paper describes the Uganda policy framework and context. Second, there is a brief discussion of the methodology. Finally, theoretical perspectives focusing on borders, integration, and sustainability are presented. The results are presented and discussed in the final section in relation to the sustainable livelihoods framework.

## 2. Theoretical Framework

### 2.1. Local Integration

There are varied conceptual dimensions of local integration. The 1951 UN Refugee convention emphasises restoring refugees to dignity and securing human rights to facilitate integration into the host community (Assembly UN General, 1951). [30] offers a more classical definition, 'a situation where the host and refugee communities can co-exist sharing the same economic and social resources with no more significant mutual conflict than within the host community'. However, local integration is often presented as an institutional response and an alternative policy to warehousing and encampment [31,32]. In this paper, local integration is conceived of as a process and outcome that facilitates opportunities for the socio-economic inclusion of refugees and peaceful co-existence with the host communities.

### 2.2. Borders and Bordering Practices

According to Donnan and Wilson (1999) [33], borders are "zones" that span geospatial features and are characterised by distinctive kinds of social, economic, and political relationships. Borders are complex and fluid and often overlap geopolitical, social, cultural, and biophysical dimensions. They serve both visible and invisible functions not limited to the geopolitical or geospatial connotations [34]. Therefore, borders have different meanings and functions beyond the "line" metaphor. First, they demarcate national boundaries and territories where sovereignty is declared or challenged. Second, they loosely demarcate ethnic, cultural, and linguistic zones. Besides, they create a regulated 'regime of mobility that requires formal citizenship instruments such as passports or any legal and internationally recognised identification documents'. Finally, borders serve as sites where movement is permitted or denied [35]. These diverse functions of borders create varied experiences of mobility and movement for different social groups. Thus, borders, rather than being fixed entities, are constituted and negotiated by different actors. They are remade and made in a context of diverse economic, political, and cultural practices that are historically defined and divergent [36].

For displaced people, borders create varied mobility and movement experiences that significantly impact their lives and livelihoods. They determine both the physical and socio-economic mobilities of refugees. For example, The 1951 Geneva Convention defines a refugee as 'a person who is outside his or her country of nationality or habitual residence; has a well-founded fear of being persecuted because of his or her race, religion, nationality, membership of a particular social group or political opinion; and is unable or unwilling to avail him— or herself of the protection of that country, or to return there, for fear of persecution' (see Article 1A (2). Thus, borders are a vital feature determining refugees' recognition, identity, and aspects of their livelihoods and welfare. In addition, refugees encounter symbolic and social borders in the places of refuge and exile. These symbolic borders in host countries and communities can enhance or undermine their livelihoods and opportunities for integration.

### 2.3. Sustainability

The paper focuses on sustainable development as defined by the Brundtland Commission, *Our Common Future*–'development that meets the needs of the present without compromising the ability of future generations to meet their own needs' [37]. Sustainability is a crucial signifier of 'good' development. Sustainability encompasses ecological, economic, and socio-political dimensions of development at local and global dimensions [38]. Increasingly, the concept has shifted from broader conceptualisation of environmental issues to livelihoods, focusing on social and economic welfare and well-being. In this way, sustainable development is perceived of as one that is inclusive, socially just, and ethically acceptable.

Furthermore, responsible development requires considering natural, human, and economic capital [39] to create a mutually supportive development for refugees and their host communities. However, the concept can be 'revigorated and reinvented' for new challenges that still elude us [38]. This paper uses the sustainable livelihoods framework to analyse the experiences of displaced people and their resilience.

The sustainable livelihoods framework (SLF) is a practical conceptual and analytical framework highlighting key factors that determine livelihood strategies and outcomes for individuals and households within different contexts. Scoones [40] emphasises how access to livelihood resources (capitals) influences livelihood strategies. Thus, SLF is a valuable tool for analysing refugee livelihoods that are preconditioned by structural and contextual factors. Refugees are highly dependent on external and humanitarian assistance for their well-being and livelihoods. However, there is limited understanding of how they strategise beyond humanitarian support. Refugees deploy their agency to survive amidst resource constraints, and this paper contributes to understanding the diverse strategies that refugees in rural settings deploy to access socio-economic spaces.

*2.4. The Uganda Refugee Policy*

National policies and legal frameworks significantly impact the rights and abilities of refugees. Uganda is home to refugees from several countries in East and Central Africa, with the largest populations coming from South Sudan, the Democratic Republic of Congo (DRC), Burundi, Rwanda, Somalia, Eritrea, and Ethiopia. The current statistics indicate that Uganda hosts 1,498,442 refugees and asylum-seekers, of which 230,101 are hosted in Adjumani district, where this research was conducted [41]. Uganda has been home to refugees since 1960, when the oldest Refugee Settlement, Nakivale in Southwestern Uganda, was established to accommodate the refugees from Rwanda [25]. Consequently, the long history of hosting refugees has shaped the policy frameworks. Uganda maintains an open border policy and a progressive asylum strategy [42]. The 2006 Refugee Act and 2010 Refugee regulations allow refugees access to land and public services; the right to work, establish a business, and own property; and freedom of movement. South Sudanese and Congolese refugees are granted refugee protection on a prima facie basis [43] The refugees can decide where they want to reside, and they have a choice over the settlements if they have established kin. Specifically, refugees are often allocated land in rural settlements for shelter and agriculture [42]

Moreover, the refugees and the host communities often have geographical proximity, share the same essential resources and livelihood activities, at times competing but often complementing one another [44]. The current protection and assistance model promotes non-encampment, self-sufficiency, and integration in the host community [45]. Besides, the settlement transformative agenda (STA) aims to foster sustainable livelihoods for refugees and host communities and create an enabling environment for refugees to live in safety, dignity, and harmony with the host communities. Therefore, implicitly the Uganda refugee policy promotes local integration as a durable solution [46]

In addition, Uganda ratified the New York Declaration for Refugees and Migrants and the Global Compact on Refugees. In March 2017, the comprehensive refugee response framework (CRRF) was launched to reinforce existing initiatives and policies to enhance refugee resilience and self-reliance. Using a whole-of-society approach, Uganda is implementing the CRRF through the Refugee and Host Population Empowerment (Re-HoPE), a strategic framework that fosters coordinated and harmonised humanitarian and development assistance to facilitate peaceful co-existence [44].

The whole society approach enhances elements of the humanitarian-development nexus by promoting progressive refugee rights, livelihoods facilitation, and integrated social services, which are crucial for integration and self-reliance [44]. The Uganda model addresses key refugee welfare constraints; however, the outcomes and impacts are mixed. For example, the high dependence on agricultural livelihoods is undermined by land scarcity, unpredictable weather, and the settlement-based aid delivery system, which restricts the pursuit of alternative livelihoods [47]. In addition, there are still challenges related to access to education, healthcare, and access to labour markets [48]. Using the ethnographic data from selected settlements in Adjumani district, this paper examines access to social and economic spaces as a pathway to local integration and sustainable livelihoods.

## 3. Materials and Methods

Ethnographic and quantitative survey methods were used in the data collection. These two methods were used to capture the diverse experiences of refugees. The ethnographic involved use of key informant interviews (KIIs) and focus group discussions (FGDs). The first phase of the research was conducted in Adjumani district, northern Uganda, in July and August 2017. The settlements visited included two new settlements, Pagirinya (est. 2016) and Nyumanzi (est. 2014), and one old settlement, Mirieyi, established in 1994. The selection criteria for the settlements were based on the date of establishment. The key informants were mainly household heads, both male and female. According to the Uganda Bureau of Statistics [49,50] a household is defined as one person or group who usually

cook, eat, and live together under the same roof irrespective of whether they are related or unrelated. In this paper, a household head refers to both female and male respondents or informants considered by their household members as the key decision-makers in their households and therefore likely to provide relevant information. The RWC, cluster, and block leaders facilitated the recruitment of the respondents.

During the first phase, 40 respondents were interviewed across the three settlements. In addition, twelve (12) key informants were selected, mainly household heads, both female and male. The key informants provided information on the individual and household livelihoods and integration experiences. Follow-up interviews were conducted with key informants who provided contextual information and periodic updates on how their situation changed up to 2019. In addition, three focus group discussions (FGDs) were conducted using a semi-structured interview guide focusing on general themes related to refugee welfare. The discussions were moderated by the author/lead researcher assisted by the research assistants. Every participant was allowed to express their views on a particular theme; the consensus was sought in cases of disagreement. In instances where it was difficult to get consensus, these were treated as independent views. FGD 1 constituted 5 People with Special Needs (PSNs), mainly elderly women. They provided information about the kind of support available to PSNs and their livelihoods experiences; FGD 2 was constituted of six host community representatives of Nyumanzi settlement. The host-community representatives provided information about interaction with the refugees, and FGD 3 constituted eight Refugee Welfare Committee (RWC) members. The refugee welfare committee discussed the assistance available to the refugees, critical displacement challenges, survival strategies, and interactions with the host communities. This information helped provide contextual information, which was also triangulated with key informant interviews. In addition, KIIs were conducted with two officials from the Office of the Prime Minister in Adjumani for policy perspectives and two representatives from humanitarian organisations on the nature of assistance provided.

During phase 2, in September 2020, a household survey was conducted using a digital research platform, Kobo toolbox. The survey was a follow-up on the issues reported during the first phase to capture the diversity of experiences. The purpose of the survey was to map the refugees' socioeconomic profile and assess the livelihood opportunities and constraints within these settlements. The survey was done in collaboration with a team of researchers from Makerere University. Four hundred sixteen (416) households were interviewed in two refugee settlements of Agogo (est. 2016) and Maaji III (est.1997). In addition, seven key informants interviews were conducted mainly with the refugee community leaders.

However, due to COVID-19 restrictions, the settlements visited during this phase were recommended by the OPM. Nevertheless, they also meet the selection criteria used during the first phase based on the date of establishment. In total, the analysis is based on sampled experiences from five settlements. It is important to note that the settlements are different in terms of tribes, economic activities, and leadership structures. These factors may influence the livelihoods, integration experiences, and nature of relations with the host communities. Moreover, the level humanitarian of assistance also varies between the old and new settlements and for different genders, age groups, and people with special needs.

In both phases of the fieldwork, it was essential to have both male and female research assistants cater to gender-sensitive aspects during the interviews. The female assistant interviewed the women, while the male assistant interviewed the men, enriching the responses provided. In addition, the research assistants were involved in the data collection and the translation from the local language *Madi* to English. In some settlements where Arabic was the primary language, refugees were hired as translators. The limitations with translation are noted, but many of the refugees could speak English and, in some instances, could interject when the translation seemed incomplete. In processing the data, the refugee informants' identities and any sensitive information are anonymised for protection.

## 4. Living with and between Borders

*4.1. Family Separation*

In Adjumani district, women and children constitute up to 86 per cent of the refugee population (OPM June 2021). As shown in Figure 1, most of the respondents were women, which also impacts the most represented views.

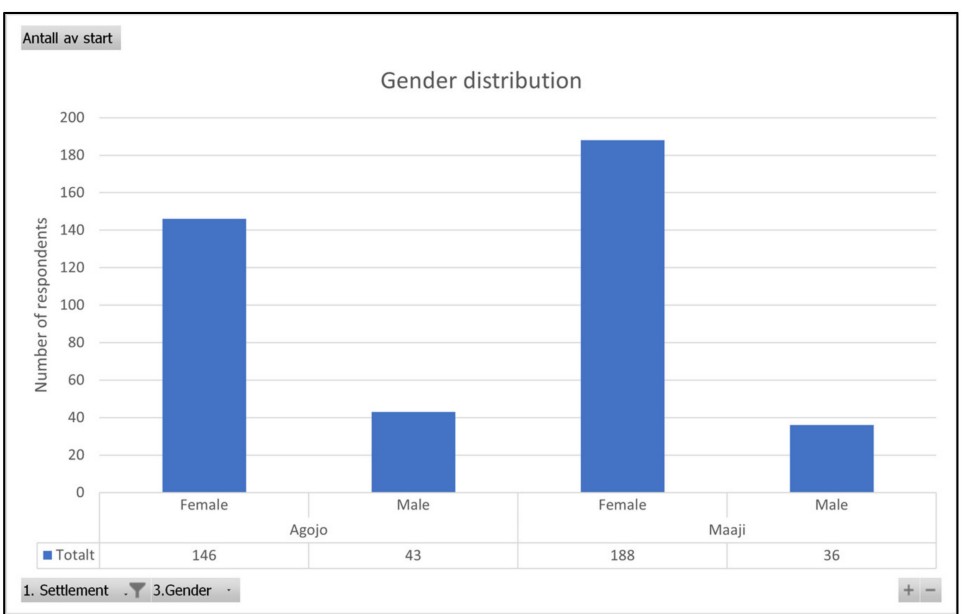

**Figure 1.** Gender distribution of respondents.

During fieldwork, it was reported that many of the children were separated from their families or orphaned. There are about 5000 unaccompanied or separated minors under the refugee welfare committee (RWC). These children are often supported as Persons with Special Needs (PSNs) in the settlement. Most of the women are separated, divorced, or widowed and single mothers. There are very few families where husband and wife live together. Most of them were separated by the conflict, or some decided to split the family, sending women and children to the settlements while the men stayed back in South Sudan. For most families, safety and access to education were prioritised for the children. Female-headed households are typical in the settlements, and foster parenting is a common strategy to support orphaned and unaccompanied minors. This quote captures the family situation in the settlements studied:

> '*Many men have moved out of the settlement to look for work, and some stayed back in South Sudan to support their families. Most men preferred to stay back in South Sudan, where they have better economic opportunities. In the settlements, there is no work for the men. For instance, me, I must look for a job for survival. I have to leave this place; I go to another place, when you just morning to sunset sit like this you cannot get something, this is not a human being you at least have to leave the place*.' (Refugee, key Informant)

Some of the refugees also reported losing contact with their relatives who stayed back in South Sudan. Most married women reported no longer receiving any economic support from their husbands who stayed in South Sudan. Some of the men are stuck in South Sudan due to the recent COVID-19 lockdown and travel restrictions. Most of the women reported that they are now focused on settling in Uganda. There is freedom of movement between settlements; however, some refugees are reluctant to transfer to other settlements for fear of losing their shelter plots and difficulty accessing land in the new settlement. Most men migrate between settlements in search of wage labour and other income-earning opportunities.

This population structure presents unique challenges regarding livelihoods, social networks, or safety nets available to vulnerable groups. For example, many mothers reported enormous care responsibilities that limit their economic participation. The school-going children are mainly dependents, and the elderly have special needs that require family support. Many elderly persons reported that they lack people to assist them with the daily chores, which was possible when they were back home in South Sudan. Thus, the physical separation of families and communities ruptured the social networks that people usually draw upon in their time of adversity. The refugee informants reported that the family and community provided material and non-material support back home, which they now lack in the settlements. However, they also reported the emergence of new social networks based on solidarity rather than kinship. Notwithstanding, these networks are considered voluntary and inadequate, given that everyone is in an equally constrained social and economic situation [26,51]. In this sense, a shrinking social space affects coping strategies and undermines livelihoods dependent on social networks.

*4.2. Language Barriers*

The language was reported as a critical barrier to local integration and compromised healthcare and education services access. According to the RWC, the inability to master the local language is a driver for high school dropout rates among refugee children. The major languages used by the refugees in the studied settlements include Arabic, Kiswahili, Dinka, and Madi. According to the Ministry of education policy, at lower primary (P1–P3), children are taught in the mother language based on the ethnicity and district where the school is located. In Uganda, over 50 tribes and languages present a unique constraint for refugee children in the integrated learning system. Most schools in the settlements use Madi, Lugbara, and Acholi as languages of instruction for lower primary schools. In addition, the parents reported that the Uganda curriculum is relevant if their children are to continue their education in Uganda. Most of the parents interviewed prefer their children to attend secondary education in Sudan, which they consider cheaper and more relevant for future employment opportunities.

Moreover, it was reported that the transition to secondary school is problematic, especially when children must return to Sudan, where Arabic is the language of instruction. However, not all parents can send their children back to South Sudan for secondary education; as a result, there are many idle youths without viable alternatives such as vocational and tertiary education or employment. Therefore, the education system does not adequately address the learning needs of culturally and linguistically diverse children.

Furthermore, some informants reported they were reluctant to seek antenatal and maternity services at the health centres due to language barriers. They reported that the midwives do not understand their childbirth choices, practices, and cultural norms and therefore resented supervised births (RWC FGD3). It was reported that some of the mothers opt for home births assisted by traditional birth attendants. Although home births were undertaken to minimise transport costs pragmatically, some facilities are about 5–10 km from the settlements. Some informants also reported that they could not access health facilities outside the settlements due to the language barrier. The government officials acknowledge the miscommunication that stems from the language barrier as a critical constraint to efficient service delivery.

*4.3. Cultural Encounters*

The host community expressed discontent with the work overload for refugee women. In some cases, refugee women did what is often considered men's work, like herding and milking cattle. The host community reported that this deviates from their cultural norms and expressed concern about the refugee women doing strenuous and danger-ous masculine tasks. Contrarily, the refugee men were considered idle and redundant. However, interviews with the refugees reveal a more complex scenario, especially from female-headed households. Many of the women in the settlements are single mothers, wid-

ows, separated, or divorced and therefore bear the full responsibility for their households, which explains their active economic engagement and 'breach' of cultural norms.

Moreover, the host community representatives raised concerns about the issues of early child marriages. Some members criticised the early marriages; meanwhile, the refugees seemed to endorse it as a cultural norm. The interviews with the young refugee mothers in the settlements also confirmed the trend of early marriages. Furthermore, the interviews with the NGOs dealing with child protection also confirmed early marriages, especially among unaccompanied minors. For example, the quotes below demonstrate these trends:

> '*Many men cannot support their wives, which is causing divorce "even these small child schoolgirls are looking for men to get something for survival. This has been there, but it has increased so much during the corona pandemic. Especially this year almost there is no even school, they are at home. So, they opt for marriage. This is a very serious issue even here in our settlement*.' (Refugee, key Informant)

> '*Schoolgirls are engaged in transactional sex. This has increased during the pandemic due to school closure*.' (Refugee leader, key Informant)

There are explicit conflicts between the cultural norms and policy about underage marriage and defilement. In Uganda, the legal age of consent for sex and marriage is 18 years; marriage for children below this age is a criminal offence. Among some refugee groups, early marriage is a common practice [52]. Thus, early marriages have significant implications for the girl-child and young mothers' health and future education and economic prospects.

### 4.4. Resource Dilemmas

Most of the refugees stay in settlements as a precondition to access humanitarian aid. Some have opportunities to grow their food, but many are highly dependent on food assistance. There is high unpredictability of the humanitarian assistance provided, and yet the humanitarian needs are persistent. Impoverishment is characterised by limited access to and control of both economic and non-economic assets. Generally, there is low purchasing power among the refugees and the host communities to support vibrant local economies. Table 1 below shows the nature of economic engagement in the settlements. Most refugees are self-employed, but the majority are unemployed and dependent on humanitarian assistance. Those self-employed are involved in tailoring, hairdressing, baking bread and pastries (chapatis, mandazi, and samosa), making handicrafts, burning charcoal, and market vendors brewing distilling alcohol. Besides, the unemployed mentioned farming as their primary occupation.

**Table 1.** Forms of employment.

| Employment Status | Number of Respondents | Percentage |
|---|---|---|
| Unemployed | 171 | 41,11 |
| Self-employed | 145 | 34,86 |
| Casual/wage labour | 25 | 6,01 |
| Private Company/business | 9 | 2,16 |
| International NGO | 5 | 1,2 |
| Other | 5 | 1,2 |
| Family business | 4 | 0,96 |
| Government employee | 4 | 0,96 |
| Local NGO | 2 | 0,48 |
| Foreign/International NGO | 1 | 0,24 |

Many of the refugees are dependent on the informal and private sectors as a source of income and livelihood. Those who have work opportunities have irregular and contract jobs with minimal social protection. Most refugees reported very low income-generating capacities due to the precarious nature of their economic engagement. Furthermore, the respondents were asked to provide information about their average monthly expenditures, as shown in Table 2. The majority of the refugees reported low expenses ranging between USD 6 and 14. These low expenses are proxy indicators of the low-income levels. The average cash grant is UGX 31,000 per refugee every month. It is important to note that refugees receive two types of humanitarian assistance: food aid and cash transfers. They can opt for either. Thus, the household income depends on the number of people in the household. The recipients of cash transfers reported higher expenditures compared to those who receive food rations. However, those who receive food rations also sell off some portions to get income to cover other household expenses.

**Table 2.** Average Monthly Expenditure.

| Monthly Expenditure | Number of Respondents | Percentage |
|---|---|---|
| UGX 20,000–50,000 | 124 | 29,81 |
| UGX 60,000–100,000 | 89 | 21,39 |
| UGX 110,000–150,000 | 92 | 22,12 |
| UGX 151,000–200,000 | 42 | 10,1 |
| Over UGX 210,000 | 38 | 9,13 |
| None | 1 | 0,24 |

The refugees were also asked to provide information about other sources of income. The intent was to find out if any of the refugees received remittances from friends or family in South Sudan or elsewhere, as shown in Figure 2 below. Unfortunately, very few refugees receive remittances, mainly from relatives, which demonstrates the refugees' limited income opportunities and precarious economic situation in rural settings.

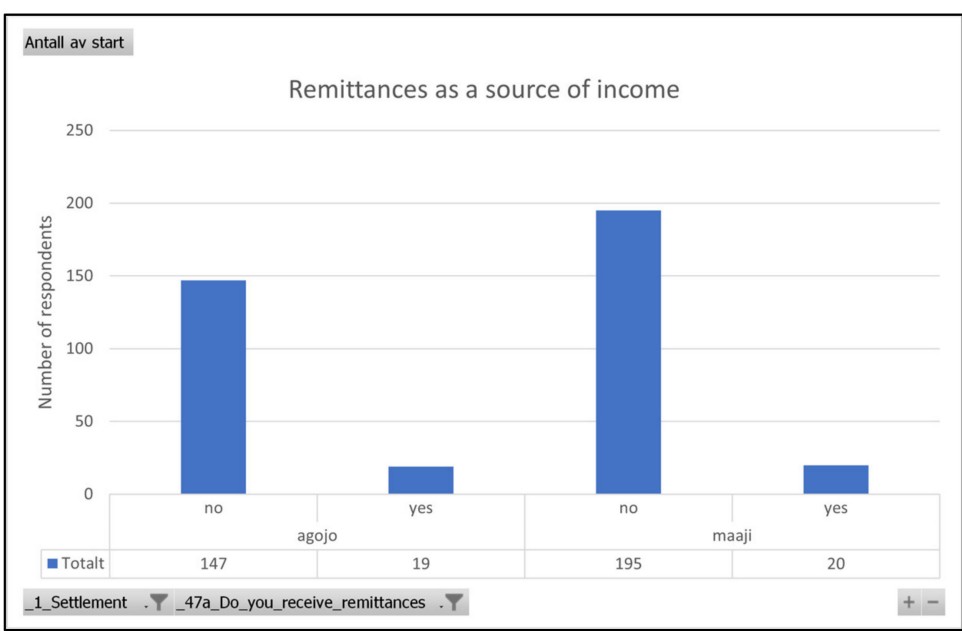

**Figure 2.** Remittances as a source of income.

Furthermore, most refugees cannot use their skills and education to support their families due to limited employment opportunities. However, despite their limited access

to livelihood opportunities, refugees deploy multiple survival strategies and desire self-sufficiency in the long term.

Immobility was reported as a significant constraint. There is limited physical and economic infrastructure due to the remote and geographically isolated location of the settlements. The long distances coupled with the lack of public transport infrastructure and high transport costs constrain the refugees' mobility beyond the settlements and settlement districts. For this reason, most of the refugees pursue sedentary livelihoods to minimise travel expenses. Some also mentioned they could not take jobs outside the settlement because of the high transport and accommodation costs. Figure 3 shows the mobility trends of the refugees. The mobility patterns also vary, as shown below. There is higher mobility in Agojo than Maaji, which could be attributed to the proximity to Adjumani Town, the primary urban centre. The estimated distance is 12 km from Agojo and 30 km from Maaji, respectively. The main reasons for travelling outside the settlement were buying or selling merchandise, seeking medical care or education, and visiting friends and family.

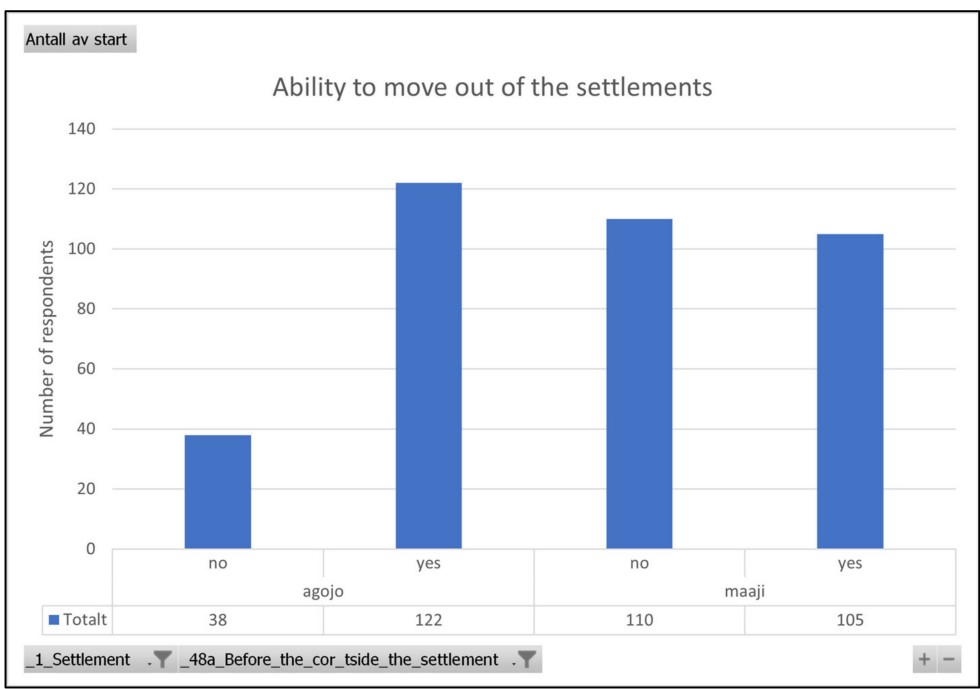

**Figure 3.** Ability to move out of the settlements.

The refugees were asked about the requirement for special permissions to travel beyond the settlements as shown in Figure 4 below. Many confirmed they did not require any special travel permissions to travel within the host district. However, those travelling outside the host district need special clearance from the Office of the Prime Minister. The OPM confirmed that this is required for protection purposes. This requirement primarily affects refugees who have engaged in trade activities and cross-border trade. For example, some traders buy fish from South Sudan and sell it in the settlements, while others buy fresh food in Uganda and sell it in South Sudan. In addition, cross-border trade is practised in settlements close to the border entry points, such as Elegu in Adjumani district. These cross-border exchanges illustrate the need for mobility as a strategy for livelihoods. For the refugees, cross border mobility has symbolic significance despite the legal dilemmas about entitlements, rights, and responsibilities associated with their refugee status. Besides, the need for socio-ritual continuities such as burial and marriages is critical for social identity. In addition, mobility was identified as an essential coping strategy amidst the fear of renewed or continued violence and insecurity in South Sudan. Thus, durable solutions should be embedded in the choices and strategies deployed by the displaced people [53,54].

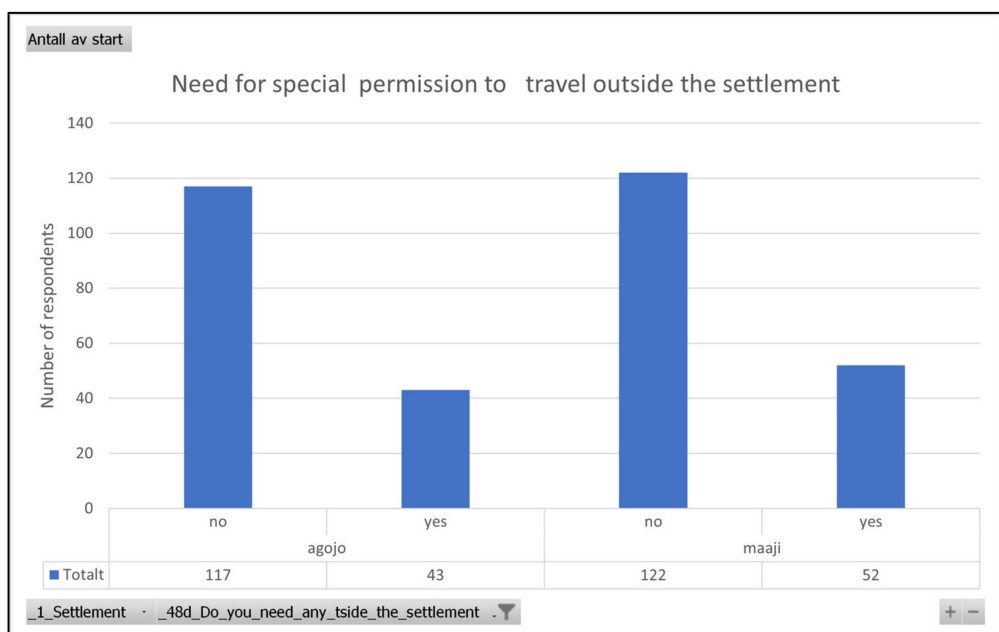

**Figure 4.** Need for special permissions to travel outside the settlement.

However, the major constraint stems from access to natural resources, as discussed below.

*Land*—providing land for shelter and farming is a crucial policy strategy for hosting refugees in Uganda. The government negotiates with the local communities to acquire the land for settlement. Land is a vital resource for refugees whose traditional livelihoods are land-based activities such as crop farming and pastoralism. Most refugees confirmed they have access to shelter plots of varying sizes, as shown in Figure 5. Those without shelter plots arrived as single persons (those without families) and are usually allocated plots that they share with other single persons. Nevertheless, there are substantial variations in the plot sizes between the settlements depending on the availability of land. For example, in Nyumanzi, the average plot is 20 × 30 m; in Mirieyi 10 × 10 m; and in Maaji 30 × 30.

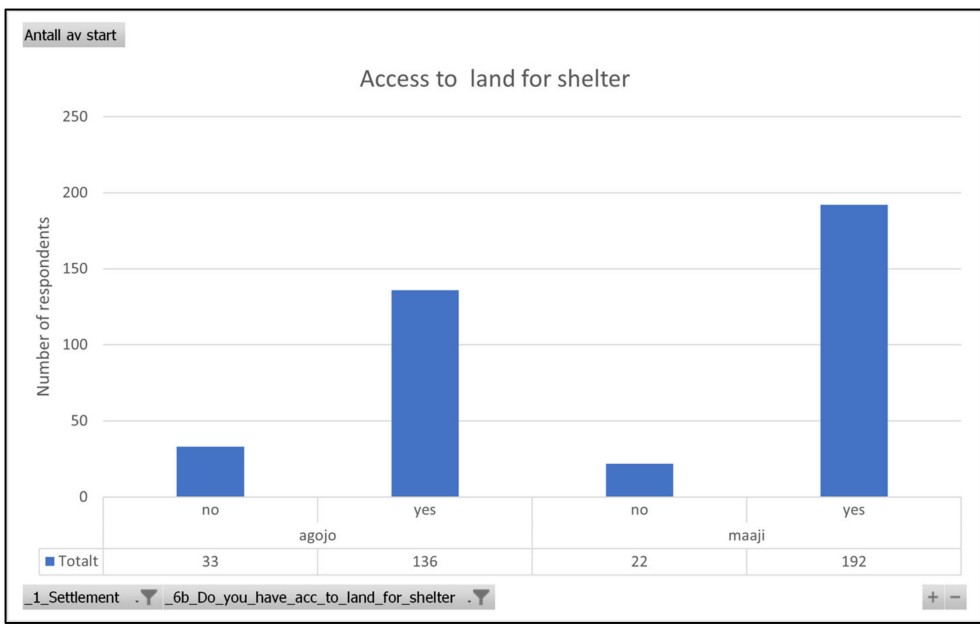

**Figure 5.** Access to land for shelter.

Regarding access to land for farming, there is considerable constraint, as shown in Figure 6. There are more refugees with access to cultivation land in Maaji than in Agojo. The refugees reported that the soils in Maaji are relatively fertile, and the rainfall pattern favours agriculture. Meanwhile, in Agojo, the land is mostly rocky, and stone quarrying was a significant economic activity. In Mirieyi, there is an acute shortage of cultivation land.

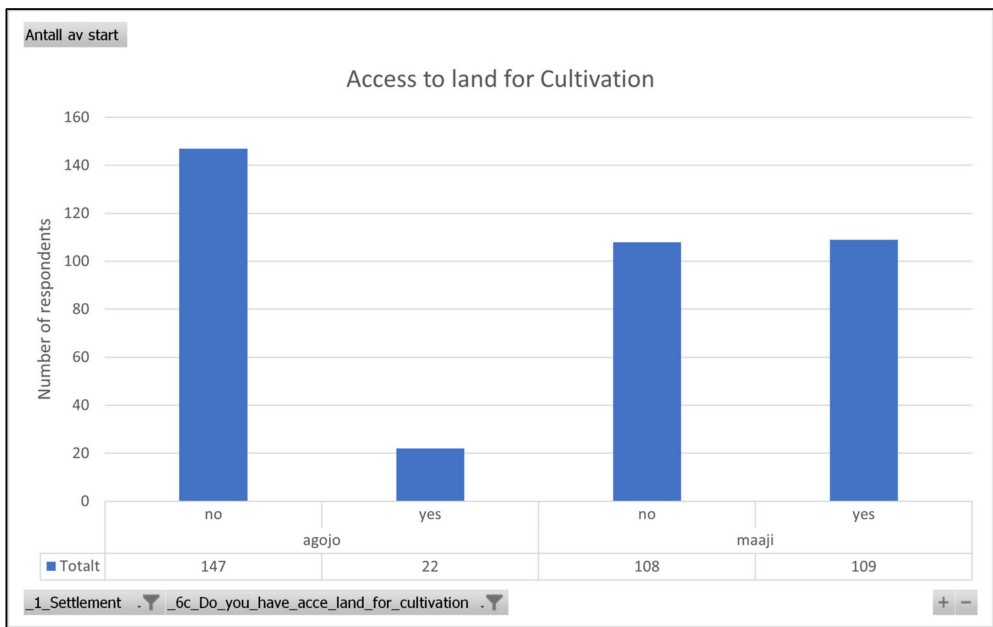

**Figure 6.** Access to land for cultivation.

Different strategies are used to acquire additional land for cultivation, such as inheriting plots from family members who have left the settlement or returned to South Sudan; hiring; or renting from the host community, as shown in Figure 7. Group farming is a common strategy deployed to address land scarcities.

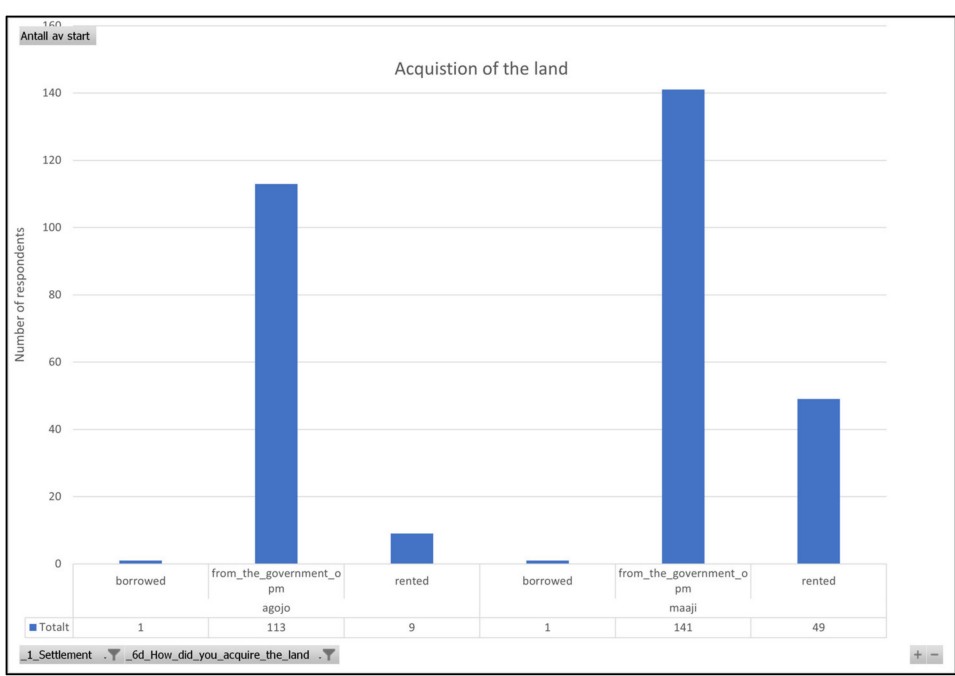

**Figure 7.** Acquisition of land.

Most land is acquired from the host community through individual negotiations and social networks; however, the challenges of access to land are growing with the increasing population. Many respondents were concerned with the shortage of land for cultivation, as narrated below:

> '*Even me when I was in South Sudan, I have been farming. Unfortunately, when I came here, I just settled in a rocky area; there is not no other alternative to do farming. I have interest in it, but there is no possibility of doing it.*' (refugee, key informant)

Additionally, in Nyumanzi, there are nomadic groups who need extensive land for grazing their livestock. Some refugees brought livestock as a critical livelihood resource and asset when they fled the conflict in South Sudan. However, there is minimal access to grazing areas in most settlements. The scarcity of land for grazing is a significant challenge for pastoralists. Therefore, most refugees only keep livestock such as pigs, goats, and poultry that do not require extensive grazing areas and avoid conflicts over stray animals destroying crops in the host communities or neighbourhoods.

Several issues were reported about the restrictions related to the use of land. For example, the landowners demand a commission on commercial activities undertaken on the land, such as farming, brick making, and charcoal burning. In addition, there are informal agreements between the refugees and landowners, including payment of rent and sharing of the agricultural produce. For instance, some refugees have commitments to give a specified percentage of the crop harvested as an in-kind payment for using the land. The quotes below illustrate other issues related to land.

> '*Land tenure period varies, and most agreements are verbal. Most refugees avoid keeping livestock due to land shortages. Most landlords prefer that tenants cultivate non-perennial crops or plants with a short maturing period to avoid 'locking' of the land and ensure short-term land agreements with the users. Most of the agreements are short term (1–3 years).*'(refugee, key informant)

> '*The host communities can give us land; however, it is not free; we rent it. The rates for renting vary based on your relationship with the landowner. It is within a range of UGX 50,000-80,000 (USD 14- 22) (1 USD = UGX 3524 -https://www.bou.or.ug/ bou/bouwebsite/BOU-HOME) for one acre per season/year. We usually sign a user agreement because if you do not sign an agreement after digging, then the landowner may retrieve his land. Most of the agreements are signed on an annual basis, but if you get a good landowner that it can be indefinite.*' (refugee, key informant)

> '*It is very easy to get land. It depends on understanding the landowner. For them (landowners) need you to be open. Tell them what you are going to do and the benefit of using their land.*' (refugee leader)

> '*The OPM lobbies for land from the host communities. The land tenure system is communal or customary tenure. There are communities that want to give land when approached by the OPM, but others refuse. For example, there are areas that were previously settled by the refugees during the previous displacement, but when the refugees returned, they declined to offer their land. The land belongs to the people, and OPM must negotiate with the landowners . . . OPM often negotiates for land through the district authorities . . . *' (local district official, key informant)

The reported land conflicts arise from contesting the purpose and nature of the land allocated to refugees. For example, there are significant challenges with tenure and user rights because individuals own the land under customary tenure. Within the customary tenure system, the land is administered by family/clan leaders. Access is based on inheritance; thus, there are limited rights and entitlements for those outside the family, clan, or community.

Many of the refugees reported that they could not invest in the land due to use restrictions. However, this was reinforced by the inherent desire to return to South Sudan, which affects the nature of investments. Therefore, the hope for return and insecure land

tenure in the settlements meant many refugees prefer short-term investments. Their life strategies span the two geospatial spaces—the place of refuge and origin—characterised as transnational livelihoods. This bordering between the present and the future often leads to precarity and uncertainty of livelihood strategies and integration.

The pressure on resources is most evident in lucrative income-generating activities such as stone quarrying, brickmaking, and charcoal burning, as narrated in the quote below.

> '*The stone aggregates trade is controlled by the host community. Refugees are not allowed to sell the aggregates to other traders. The host community are middlemen, and this has created some exploitation. For example, buyers from the host community buy a heap for UGX 25,000 (USD 6.8) and sell at UGX 75,000 (USD 20.5). If refugees try to sell the aggregates to other buyers, they are arrested, and the stone aggregate is confiscated. They need to obtain permission from the Host community to sell to others*.' (refugee, key informant)

The refugees who can afford to rent or hire cultivation land privately negotiate with landowners in the host communities. Thus, the nature of land-use agreements is dependent on the relationship with the landowner. There are no specified terms, and therefore land use agreements vary widely. However, the breach of land use agreements was reported as a regular occurrence. To mitigate conflicts, the OPM and the RWCs are exploring options for formal and written agreements, as stated in this quote:

> '*To manage the pressure on grass and woodlots, there are local bylaws to seek to mitigate harassment and exploitation of refugees. Those (host community members) that breach these laws are punished. Most conflicts are individual-based and resolved by the local leaders*.' (refugee leader, key informant)

Overall, access to resources is constantly negotiated in the face of various restrictions and scarcity. To access resources, especially land, refugees negotiate new forms of identity and membership in the new communities to access the entitlements embodied with the membership. Thus, refugees cross the physical territorial borders and encounter socio-cultural, political, and economic borders that enable or restrict access to socio-economic resources. The constraints reported above demonstrate the dilemmas that refugees encounter in creating meaningful and sustainable livelihoods. The lack of physical and economic assets, particularly land in rural areas, is a formidable barrier to sustainable livelihoods and local integration.

### 4.5. Host Community Relations and Interactions

The Uganda model focuses on integrated service delivery for the host community and the refugees. These services include education, healthcare, water, and markets. It was reported that this has significantly increased the interaction between the host communities and the refugees. Most refugees reported amicable and cordial relations with the host communities except for a few conflict areas. Notably, there is a long history of displacement, migration, and settlement between northern Uganda and Southern Sudan [9,55,56]. Moreover, several initiatives encourage peaceful co-existence, such as joint training programmes, cultural galas, and dialogue meetings between the refugees and host communities. The positive interactions reported include:

*Trade and informal exchanges*—the economic benefit from the refugees was the significant increase in demand for goods and services generated through renting of land/property, improved services, and infrastructure supported, through humanitarian and development initiatives. There is significant interdependence, for example, the host community buys aid supplies and food rations from refugees in exchange for charcoal and firewood. The host community also sells fresh food such as maize, okra, sesame, cassava, and groundnuts in exchange for clothes and other aid goods from the refugees. Thatch grass, firewood, and timber are bought from the host community.

Meanwhile, some people from the host community come to the settlement searching for wage labour, for example, building houses and digging pit latrines. At the same time, some refugees cultivate gardens for the host communities as a mutual exchange for the land. Furthermore, collective farming is practised in some settlements whereby the host community contributes the land. The labour is provided by both groups, while NGOs and charity organisations provide farm inputs and seedlings. The crop harvest is shared equally among the group members. Crucially, some refugees hire or borrow agricultural land from the host community.

Some refugees and the host community members collaborate through saving and cooperative credit organizations (SACCOs). Besides, the development initiatives from some aid organisations and NGOs support joint livelihood activities and projects, strengthening community relations and further expanding the social networks for the refugees that enable access to other resources. Furthermore, some members from the host community are also hired during the food distribution days to off-load aid goods from the delivery trucks.

*Social and cultural interactions*: positive relations were also reported through sharing churches, religious, cultural, and sports activities. In addition, extra-curricular activities such as interschool debates and sports competitions foster positive relationships among school children. Youth leaders are actively involved in organising joint dialogue meetings and sports, especially football. Community recreation centres, boreholes, and marketplaces are meeting points for the refugees and the host community. Intermarriages and joint cultural celebrations are positively appraised. These interactions and exchanges enhance the livelihoods and welfare of both the refugees and host communities.

The negative interactions reported include:

*Ethnic and inter-community violence*: there were reports of intra and intercommunity violence, revenge killings, and kidnappings among the refugees, which are an overflow of the tribal conflicts and the political faction in South Sudan. Inter-tribal conflicts were reported in some settlements that have both Dinka and Nuer. Refugees from ethnically homogenous communities have encountered tension and conflict with other tribal groups when they meet in the settlements. Some refugees also reported fear of being tracked by rival tribal groups from other settlements. In one settlement, the RWC reported an incident in which conflict between the host community and the refugees resulted in the death of 20 people. In addition, conflicts between school children often spill over and trigger fights between the communities.

*Share of aid resources*: the host community representatives expressed their dissatisfaction with the aid ratios policy. They consider the refugees as more privileged, which is a source of tension. According to the community representatives, the 30–70 per cent ratio of access to aid and social services is unfair. The service delivery ratio means that in terms of access to social services such as schools and hospitals, the host communities get a quota of 30 per cent of the total vacancies in the school. In contrast, the refugees get 70 per cent of the vacancies. The host community argues that this is insufficient compensation for their land. The host community representatives argued for equitable access to social services. In addition, the food rations provided to the refugees are contentious. The host community argues that they are equally constrained, comprising the provision of humanitarian assistance when there is a need. For instance, in 2016, there was a severe drought, but the host communities did not get any food assistance. These non-inclusive aid policies undermine peaceful co-existence and livelihoods.

*Breach of boundaries and borders*: the trespassing between the villages and settlements has been a source of conflict. It was reported that refugee youth often invade the gardens of the locals to get food during periods of food scarcity in the settlements. There were reports of the youth stealing maize, fruits, and other food items from the gardens of the host communities. The host communities reported the need to devise strategies to protect their crops, which has also become a source of conflict. The host community also reported alcohol abuse and witchcraft as critical constraints to positive interactions. Besides, the lack of burial grounds is a contentious issue for the refugees and the landowners. According

to the key informants, there are no designated burial sites within the settlements. For instance, there was a recent grave on their shelter plot in one of the homes visited. As a customary practice, the refugees prefer to repatriate the dead back to South Sudan, but many cannot afford the expenses involved. However, the landowners were weary of the spiritual consequences of burying strangers on their ancestral land. Besides, some landowners reported that they had offered land for burial, but they never received any compensation from the government. Thus, this has been a source of conflict, yet the refugees desire to give their loved ones a befitting burial.

Furthermore, the host communities reported the competition over common pool resources as a critical dilemma. For example, firewood and thatch grass are used by both the refugees and the host community. The refugees reported the lack of construction materials such as thatch grass and poles for construction. However, they expressed concern that the host communities were unwilling to give free grass or sell it at a reasonable price. For the host community, the critical concerns arise from over-harvesting and the timing of harvesting the grass. The locals say it is best to harvest the grass in November when it is matured; however, the refugees harvest grass prematurely, affecting its regeneration and availability. Besides, fencing the homestead is a cultural practice among the Dinka. However, according to the host community, this is a new practice that is causing massive deforestation and environmental degradation (FGD 2). Finally, water is a vital a shared resource between the refugees and the host communities. However, there are challenges of contested ownership and usage of some water points outside the settlements. For example, in one settlement, it was reported that the refugees use the waterholes for bathing and watering animals, resulting in contamination of drinking water sources and leading to conflict and hostility between refugees and the host communities.

## 5. Discussion—Fostering Sustainable Livelihoods for Refugees and Their Host Communities

Understanding refugees' livelihoods strategies is critical in fostering social and economic interdependence within and between communities and restoring social networks [57]. This paper highlights the socioeconomic opportunities and constraints that refugees encounter through interaction with their host communities. Most studies focus on livelihoods in situations of 'active' political instability, humanitarian emergencies, and chronic conflict situations, where the role of violence and the political economy of war and destruction of livelihood systems are emphasised [58–61]. However, limited studies within 'inactive' conflict and humanitarian contexts focus on protracted displacement, vulnerability, and livelihoods [62–64]. In this section, the paper discusses how protracted displacement into new contexts strengthens or impairs the refugees' resilience and livelihoods and their host communities. Figure 8 below is a modified version of the livelihoods framework that considers the effects of protracted conflict and displacement and how these influence refugee livelihoods. Displacement significantly affects access to vital resources because it entails relocation into new environments with different social, political, economic, and environmental conditions.

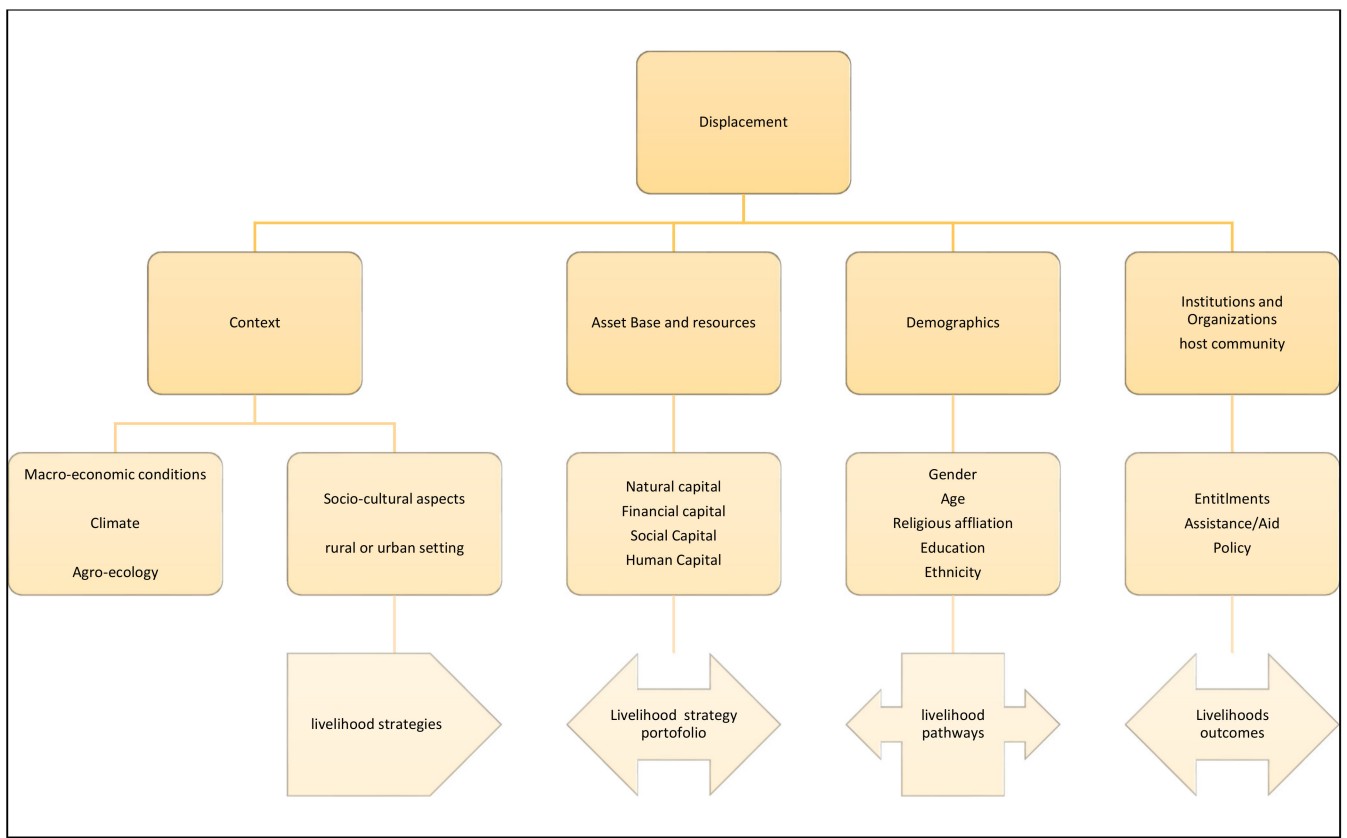

**Figure 8.** Modified sustainable livelihoods framework.

People are forced into new socioeconomic contexts that may influence the nature of their vulnerability, access to resources, institutions, and organisations. For refugees, the relocation into a new territory either narrows or expands the resource base depending on the social context and the institutional and policy frameworks of the host country and community. In addition, refugees move into new contexts where their rights and entitlements are restricted or regulated in a way that affects their agency and ability to strategise. For example, refugees fleeing from violence and political upheaval search for safe places where their security is not threatened. However, in the new contexts, refugees must negotiate for access to physical, economic, and social resources such as land, water sources, grazing lands, woodlots, and work opportunities critical for rural livelihoods. The findings of this study demonstrate that refugees are diverse and pursue multiple livelihood strategies. For instance, the refugees with rural and agricultural backgrounds were more adaptable than the former city dwellers who struggled to adjust to the new context. Besides, refugees pursue diverse livelihood trajectories based on their socio-cultural, demographic, and economic conditions. For instance, refugees with higher levels of education have higher career ambitions and aspirations and therefore opt for secondary migration either to the cities or await resettlement in the third countries.

Furthermore, the rupture and emergence of new community networks and structures is a crucial feature of displacement [24,65]. There are diverse social and interpersonal ties, collective action, and new patterns of social organisation such as Refugee Welfare Committees (these are elected representatives from the refugees who liase and coordinate with Office of the Prime minister and aid agencies on all matters concerning the refugee welfare within every settelement). However, the disruption of previously established mutual support networks is a significant livelihood constraint. Most refugees rely on informal networks of mutual support due to family breakdown and alienation caused by displacement. Moreover, different demographic groups, for example, unaccompanied minors;

single, widowed, or separated women; the disabled; and the elderly, use diverse coping mechanisms and have different contextual experiences that enhance or undermine their networks and social capital. Additionally, governments, institutions, and organisations often use borders as geopolitical and administrative units to define the eligibility of status of refugees and the right to protection and humanitarian assistance. Specifically, borders define the limits for inclusion or exclusion and the type of membership and entitlements that members can claim or are denied in a specific context.

Displacement significantly erodes human capital. When people flee from conflict, they abandon their income, jobs, and livelihoods, significantly impacting their resource base. Furthermore, when refugees move into new contexts, they risk losing their skills in a new system because of incompatibility of education systems and regulations for specific occupations and institutionalised employment and recruitment practices. For example, the Ministry of Education in Uganda requires that teachers be licensed and registered to teach in public schools. Therefore, most refugee teachers serve as class assistants because their qualifications are not fully recognised [66]. Additional challenges relate to children's education in a new educational system, especially when they anticipate returning to their home country. For example, Uganda's national integrated early childhood development (ECD) policy (March 2016) promotes mother-tongue/local language in lower primary schools. Thus, dealing with a new curriculum and foreign language was reported as a critical barrier that threatens educational outcomes and impairs human capital.

Gender, age, and ethnicity also influence access to resources, livelihood strategies, and outcomes. Notably, the cultural and economic similarities between the refugees and the host communities have significant implications for access to resources. Some refugees acknowledged that common language, shared cultural norms, and traditions between some ethnic groups are crucial to accessing social networks and livelihood opportunities. For example, the South Sudanese refugees and host communities in Uganda are predominantly farmers and pastoralists. Therefore, the land is a vital resource for production systems and commercial activities for both groups. However, the similarity of livelihoods creates increased conflict and pressure on the limited land and natural resources. For example, the host community reported conflict over grazing lands and cattle raiding. In addition, there were reports of animal theft due to the shared grazing areas. Besides, the host communities often brand/tag their animals for easy identification which the refugees consider abominable. Therefore, such cultural differences pose a threat to peaceful co-existence and undermine the prospects of local integration. The host community seeks to preserve their identity, culture, and ways of life and so do the refugees. These differences in the ways of life, diets, and cultural practices, however subtle they appear, often reinforce borders and boundaries that threaten access to socio-economic spaces and therefore undermine the prospects of sustainable livelihoods and local integration.

Refugees also encounter new social and political structures that affect their rights and entitlements to resources. Generally, refugees are protected under the 1951 Geneva Convention legal framework that demands host states provide or facilitate access to resources such as employment, education, housing, and healthcare. However, in Low-Income Countries (LICs) such as Uganda, the provisions are limited due to the state's insufficient financial capacity. Thus, refugees need to negotiate at the local levels to access land, job opportunities, property, and education, often governed by social institutions. Fundamentally, the lack of physical and economic assets is a significant deterrent to sustainable livelihoods. Therefore, the interaction with the host communities is vital to expanding the resource and asset base for refugees. The whole-of-society approach is an inclusive strategy supporting humanitarian and development initiatives for both refugees and their host communities [67] The constraints discussed above highlight the need to expand and strengthen both groups' asset base and human endowment. However, there is a need for further research to understand the economic function [68] of both refugees and their host communities in order to create vibrant rural economies. Thus, future research is needed to

provide knowledge on how to strengthen the capacities of refugees to pursue productive lives as a pathway to self-reliance and better integration in their host communities.

Essentially, displacement and exclusion from a geographical territory create significant economic, material, and cultural losses. Moreover, the resultant deprivation of access to assets and resources is a crucial driver of impoverishment. In addition, the remote location of the settlements aggravates the physical and economic immobility. Therefore, many refugees pursue passive livelihood strategies often characterised by low and limited economic returns. Thus, the precarious nature of the economic profiles and activities of the refugees are unsustainable in the long term, with many refugees awaiting opportunities for secondary migration. Overall, the experiences of refugees presented in this paper demonstrate refugees' agency towards improving their livelihoods despite the socio-economic constraints they face.

## 6. Conclusions

The findings of this study demonstrate how the structural conditions in the rural areas undermine and threaten refugee livelihoods and integration. The preexisting deprivation and social exclusion in rural areas is a critical challenge [12]. Rural areas are often characterised by social deprivation, which amplifies the isolation and social exclusion among refugees. Thus, the placement of refugees in rural areas fails to address the main challenges they experience in everyday life and intensifies the social and economic deprivation presented in this paper. Most governments have preferential resource allocations for rural areas to make them economically and socially attractive. For instance, in Uganda, aid resources are channelled to support refugee-hosting districts; however, the impacts are minimal. The striking absence of public infrastructure, the lack of satisfactory employment for working adults, and children's poor education opportunities make the rural areas unattractive. Furthermore, the settlement-based assistance policy thwarts the efforts for integration and economic mobility. The systematic concentration of refugees in settlements leads to ethnic clustering that promotes residential and social segregation [12] There are two sides to this concentration and clustering; on the one hand, it affords refugees vital networks from within but fails to bridge networks with the local communities. On the other, it significantly reinforces discrimination, marginalisation, and poverty.

Demographic change in rural areas presents both opportunities and constraints. For example, the influx of refugees in rural northern Uganda is creating conflict over natural resources. As a result, there is significant pressure on social services such as water, education, and healthcare services. Furthermore, the current land allocation policy is creating an enormous strain on land resources leading to land degradation due to over-harvesting of wood resources and vegetation for firewood and construction. These practices are environmentally and economically unsustainable for nature-based livelihoods.

Moreover, the restrictive land rights for refugees significantly impact their ability to integrate and undermines their economic engagement as discussed above. Besides, the demographic change in terms of the population structure creates diverse socio-economic needs. For example, the demand for education services has increased, and at the same time, the opportunities for employment and meaningful economic engagement are declining. Therefore, the prominent labour surplus in rural areas is a reservoir for secondary migration. Therefore, rural transformation processes should enlarge economic opportunities that engage the dynamic and youthful labour.

Mobility is a critical strategy for refugees to access physical and economic security as well as social goods. However, refugees' economic spaces are restricted due to limited access to economic resources such as land, productive assets, and labour markets. In the current form, durable solutions provide a narrow conceptualisation of solutions to protracted displacement. Local integration, voluntary repatriation, and resettlement are often considered as mutually exclusive, permanent, inflexible, and rigid, and hardly capture the realities of displaced people. Nevertheless, refugees use their agency to harness the opportunities within the different durable solutions and navigate immobility constraints.

Therefore, the classification of durable solutions as different options is superficial and limiting.

Negotiating for entitlements and access to resources defines whether people can integrate or pursue sustainable livelihoods in the new societies. Navigating the rules, norms and values that govern access to economic and social spaces are a crucial constraint to integration and sustainable livelihoods. Uganda's open border and progressive asylum policies have significantly reduced the barrier to navigating the geopolitical borders, making it an attractive destination for refugees. Freedom of movement, right to work, and choice of residence expand the opportunity space through enabling access to economic opportunities and social networks. However, there are pressure points regarding access to economic resources, livelihoods, and jobs that constrain the capacity for self-reliance, as demonstrated by the experiences of the South Sudanese refugees in Adjumani District in Northern Uganda.

**Funding:** This research was funded by The Research Council of Norway, grant number 288788.

**Institutional Review Board Statement:** Ethical review and approval were waived for this study, due to the COVID-19 restrictions which complicated the process of acquiring formal ethical approval. However, research clearance was obtained from Office of the Prime Minister (OPM) after vetting and approval of the research proposal. Besides, in the settlements there was further vetting of the proposed research by the Refugee Welfare Committee (RWC) who were also the gate-keepers to the respondents. Participation was voluntary, and respondents were allowed to withdraw from the interview without stating any reason. Besides, they could skip questions they were not ready to answer. There is no direct reference to the respondents, fictious names were used to protect their identity.

**Informed Consent Statement:** Informed consent was obtained from all the participants involved in the study.

**Data Availability Statement:** Not applicable.

**Acknowledgments:** I acknowledge the technical and logistical support received from Titus, Jogo (OPM,) Paul Mukwaya, and Gabiri Geofrey, who assisted in the data collection under the most challenging circumstances due to the corona pandemic. I also acknowledge the support received from my collegues Anne Margrethe Brigham and Pia Piroschka Otte. I would also like to thank my research assistants Kakayo Liz and Opiku Daniel. Lastly, I extend my sincere gratitude to my key informants and respondents for kindly sharing their experiences.

**Conflicts of Interest:** The authors declare no conflict of interest. The funders had no role in the design of the study; in the collection, analyses, or interpretation of data; in the writing of the manuscript; or in the decision to publish the results.

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
