# Peer review of "Local Integration as a Durable Solution? Negotiating Socioeconomic Spaces between Refugees and Host Communities in Rural Northern Uganda"

_sustainability, doi:10.3390/su131910831_

Round 1
Reviewer 1 Report
I enjoyed this paper, which I think is an original and useful contribution to the broad challenge of looking at refugee communities through a sustainability lens. I certainly support its publication but have a few issues I would like the author to consider.
First, and most significantly for me, the paper should 'talk up' the sustainability issue a bit more, not least given the title of this journal! In particular, the Sustainable Livelihoods Framework needs clear and explicit introduction in the opening sections of the paper if it is to be used. Otherwise, it comes too much ‘out of the blue’ on page 12. The section that uses this in analysis (pp.12-3) then needs to be very clear about what SLF brings out / discovers / emphasises. Clearly show its value / usefulness here.
My second main concern lies with Section 4. This is a fair discussion of many of the key findings but I think the paper could usefully end with some short and sharp conclusions that the reader can clearly take away from the paper.
Third, you start with the rural emphasis and certainly for me as a European the idea of refugees being notably present in rural areas is somewhat unusual. I'm wondering then if you could perhaps make more of the challenge of keeping refugees in rural spaces – in the UK there is a tendency for them to move to where others are concentrated for many reasons but clearly these communities seem well establishedin rural Uganda.
Fourth, from a sustainability point of view I'm thinking about inherent(?) tensions between creating sustainable communities and the idea (misplaced?) that refugee communities are inherently temporary? Yet I also note how links back and forth e.g. to S Sudan, as clearly noted on p6., seem vital. You could reflect on this as it undermines any singular sense of place / home perhaps?
Some other small points:
p.6. Join ‘Nonetheless…’ paragraph to previous one as it is on the same point.
p.7 Why are there concerns about what women are doing, whether work or early marriage? Is it from a welfare perspective or almost exclusively because it breaks cultural norms?
p.7. By ‘legal age of consent’ do you mean for marriage or for sex?
The land use issues noted certainly challenge the idea of the refugees being well integrated into Ugandan society, as a sustainability focus would (arguably) require…
p.11 – ‘they did not get any food assistance’.
p.11 – ‘invade the gardens’ not ‘evade the gardens’.
Use ‘the’ a bit too much – sometimes not necessary – but very minor point!
p.11 – ‘refugees harvest grass prematurely’ sounds better.
Author Response
Dear Reviewer, I am very grateful for the feedback. I have now effected the revisions as suggested. Please see the attached file for the specific responses.

Reviewer 2 Report
The paper is very interesting and investigates how the demographic shift shapes the interaction between refugees and host communities and the access to socio-economic space in the rural communities in northern Uganda. Especially investigates the experiences of refugees in rural settlements in the Adjumani district where a significant number of refugees from mainly neighboor countries were concentrated. The issue has national/local interest but at the same time do not leave no one's reader indifferent.
The authors focused on the specific country and district, whithout to connect the issue with the international changes and the refugees waves in the rest world.
The paper have qood points but and very important weaknesses.
1. The Introduction section must be re-organized (for example, in three-four different points of Introduction, is repeated the research purpose).
2. In my opinion, the 'Theoretical Framework' must be re-numbered (to be section 2 and not subsection 2.1). The parts of this section must be numbered too (Local Integration as 2.1 and Borders as 2.2). The subsection 1.1 ( The Uganda Refugee Policy) must be transfered to the Theoretical framework as sub-section 2.3.
3. The 'Materials and Methods' must be re-numbered too (as section 3 and not as subsection 2.2).
-Why were selected only 12 key informants (household heads)? The population of refugees in this district is over 200.000.
- How worked the researchers with the three focus group discussions (FGDs), which method they followed? Please describe.
-The sample is representative?
- Why the sample of 416 households were from only two refugees settlements?
4.Why the authors do not presenting in the rest manuscript the quantitate results from the survey?
5. The sections 3 and 4 are presenting the results, discussion and conclusion section. In my opinion these sections must be re-writing and re-organized. The results which were presented are mentioning mainly to info that the key informants gave and by the FCDs in less frequency. Quantity results from survey are missing .
6. The absence of a serious Discussion section is obvious also. There is not results comparison with similar researches international oriented, they are not presented the research limitations and the future research.
7. The conclusion section is missing too. The authors describe and present a situation without strong arguments (except some refugees’ opinions and aspects who described). In the last sections missing citations or strong arguments from many paragraphs, in these cases the authors opinions are looks like to based on air.
8. The references must be enriched significant.
9. There are many repetitions in many points of the manuscript.
Author Response

(The authors gave the same response as above.)

Round 2
Reviewer 2 Report
No comments